# Risk Factors of Complications from Central Bisectionectomy (H458) for Hepatocellular Carcinoma: A Multi-Institutional Single-Arm Analysis

**DOI:** 10.3390/cancers15061740

**Published:** 2023-03-13

**Authors:** Atsushi Nanashima, Susumu Eguchi, Toru Hisaka, Yota Kawasaki, Yo-ichi Yamashita, Takao Ide, Tamotsu Kuroki, Tomoharu Yoshizumi, Kenji Kitahara, Yuichi Endo, Tohru Utsunomiya, Masatoshi Kajiwara, Masahiko Sakoda, Kohji Okamoto, Hiroaki Nagano, Yuko Takami, Toru Beppu

**Affiliations:** 1Department of Surgery, University of Miyazaki Faculty of Medicine, Miyazaki 889-1692, Japan; 2The Kyushu Study Group of Liver Surgery, Nagasaki 850-0015, Japan

**Keywords:** central bisectionectomy, hepatocellular carcinoma, perioperative risk assessments, bile leakage, liver failure

## Abstract

**Simple Summary:**

This study aims to clarify the specific perioperative risk factors and short-term patient prognosis after central bisectionectomy (CB) for hepatocellular carcinoma (HCC). The specific operative risk factors for liver failure and increased blood loss in 142 HCC patients undergoing CB (H458) were tumor involvement in segment 1, tumor size exceeding 10 cm, and compression of the main vasculature. The three-year survival after CB was favorable and curable under precise preoperative simulations, elaborate techniques, and management. Large tumor size, surrounding tumor involvement, or compression to major vasculatures, and the related iBL were independent risk factors for severe morbidities in CB. These results showed the limits of up-front CB, which is useful information for deciding other novel non-operative treatments for conversion surgery.

**Abstract:**

Background: This study aims to clarify the perioperative risk factors and short-term prognosis of central bisectionectomy (CB) for hepatocellular carcinoma (HCC). Methods: Surgical data from 142 selected patients out of 171 HCC patients who underwent anatomical CB (H458) between 2005 and 2020 were collected from 17 expert institutions in a single-arm retrospective study. Results: Morbidities recorded by the International Study Group of Liver Surgery (ISGLS) from grade BC post-hepatectomy liver failure (PHLF) and bile leakage (PHBL), or complications requiring intervention were observed in 37% of patients. A multivariate analysis showed that increased blood loss (iBL) > 1500 mL from PHLF (risk ratio [RR]: 2.79), albumin level < 4 g/dL for PHBL (RR, 2.99), involvement of segment 1, a large size > 6 cm, or compression of the hepatic venous confluence or cava by HCC for all severe complications (RR: 5.67, 3.75, 6.51, and 8.95, respectively) (*p* < 0.05) were significant parameters. Four patients (3%) died from PHLF. HCC recurred in 50% of 138 surviving patients. The three-year recurrence-free and overall survival rates were 48% and 81%, respectively. Conclusions: Large tumor size and surrounding tumor involvement, or compression of major vasculatures and the related iBL > 1500 mL were independent risk factors for severe morbidities in patients with HCC undergoing CB.

## 1. Introduction

The safety and effectiveness of anatomical or major hepatectomy for hepatocellular carcinoma (HCC), even for large and multiple cancers, has improved worldwide [1]. Among them, anatomical central bisectionectomy (CB) or mesohepatectomy for large HCCs occupying all of Couinaud’s segments 4, 5, and 8 have been selected as options. These are technically demanding, highly specialized procedures carried out by expert liver surgeons, because CB would be considered as an alternative to extended hemi-hepatectomy for selected patients with limited liver functional reserves or future remnant parenchymal volumes [2,3,4]. When the liver functional reserve is well preserved for trisectionectomy, the remnant liver should be safely secured by an optimal liver function evaluation or resected volumes, perioperatively. While considering the functional liver for adjuvant or additional treatments for HCC, up-front CB may be recognized as a standard optimal hepatectomy. Historically, CB for liver cancer was first successfully performed by McBride and Wallace in 1972 [5]; additionally, Hasegawa et al. discussed CB as a central bisegmentectomy in 16 patients with HCC and liver metastasis in 1989 [6]. In the latest comprehensive notation for hepatectomy, as per the “New World” nomenclature established in 2021, the term of CB is used as a type of hepatectomy and is indicated as H458 or H4581 [7]. Recently, a comparison between CB and other major hepatectomy procedures has been reported, and the surgical outcomes and patient prognosis in patients undergoing CB were not significantly different [3]. However, the details of the functional or anatomical risk factors or limitations of clinical radiomic selection have still not been fully clarified, particularly in a large or invasive HCC. A disadvantage of CB procedures is that it often leads to lethal patient outcomes due to the limited remnant volume of liver parenchyma [8]. In the event of a larger HCC compressing the surrounding major vessels, the difficulty of transection procedures around the hepatic veins (HVs), caudate lobe, inferior vena cava, and hilar portal (Glissonean) pedicle increases. Involvement of the intra-hepatic major vasculature accompanied by the rapidly progressive HCC influences the operative difficulty and critical adverse events [8], which are expected to be more difficult in comparison with progressive HCC occupying the hemi-liver. To estimate the operative risk related to post-hepatectomy complications, a larger number of cases, including other liver tumors, might be necessary. Cases of intra-hepatic cholangiocarcinoma (ICC) or metastatic liver cancers (MLC) undergoing CB have not been well observed because ICC is a rare condition and MLC patients tend to undergo limited resection in our institutes. Thus, we focused on HCC by considering homogenous subjects for better understanding, even though the number of subjects was limited. Furthermore, not only the surgical risk, but also the oncological point of view in centrally located HCC after CB was another important issue to decide the operative indication. Our study group separately analyzed these two issues because the number of analyzed subjects was too high. In this study, we focused on clarifying the surgical risk parameters relating to postoperative morbidity and mortality first. Taken together, our group hypothesized that intra- or extra-hepatic major vascular injury due to centrally located and large HCC was specifically associated with postoperative severe morbidities and mortality or short-term post-hepatectomy survival. If serious risk factors can be identified preoperatively, effective systemic chemotherapy is first applied as a neoadjuvant strategy to reduce or avoid HCC progression in the recent era of new chemotherapy [9]. For centrally located HCC, the real issue lies in how we can provide a safe parenchymal resection with acceptable functioning liver remnant without postoperative liver failure. Because the number of CBs for HCC was very limited in each institute, we firmly believe that the multi-institutional collection of data with a large number of patient data and the cooperated analysis should be necessary.

To clarify our hypothesis, the Kyushu Study Group of Liver Surgery (KSGLS) aimed to conduct a retrospective cohort study by collecting surgical data and analyzing 171 patients undergoing CB for HCC between 2005 and 2020 from all 17 expert liver surgery institutes in a single-arm setting when the present operative procedures under open laparotomy might be established. Using a regional multi-institutional analysis, this study investigated the specific and detailed perioperative risk factors and novel information for defining the optimal operative indications for CB. In addition to clarifying the predictive risk parameters for severe postoperative complications, we would like to explore the borderline limits of challenging up-front CB and indications of other optimal non-operative novel treatment modalities based on the present results.

## 2. Materials and Methods

### 2.1. Patients, Research Collaborators, and Ethics

All patients with primary HCC who initially underwent CB between January 2005 and December 2020 in 17 of 20 expert institutions for liver surgery belonging to the KSGLS [10], consisting of the Department of Surgery at the University of Miyazaki, Nagasaki University, Kurume University, Kagoshima University, Kumamoto University, Saga University, National Hospital Organization Nagasaki Medical Center, Kyushu University, Saga Medical Centre Koseikan, Oita University, Oita Prefectural hospital, Fukuoka University, Kagoshima Kouseiren Hospital, Kitakyushu City Yahata Hospital, Yamaguchi University, National Hospital Organization Kyushu Medical, and Yamaga City Medical Center were enrolled for the purpose of providing surgical data in the present study. This study was approved by the ethical committee of the University of Miyazaki and institutional ethical approval was obtained (approval number #O-0962, 9 June 2021). Informed consent for data collection was obtained by the opt-out procedure for one month; however, there was no disclaimer. Based on this principle, the approval from the institutional review board of each institution was obtained from the other 16 institutes mentioned above by collecting data until 17 February 2022. Subsequently, the case report forms of 171 patients who underwent CB were collected and analyzed by the principal investigator. The other three institutes of the KSGLS did not have data regarding central hepatectomy during this period; nevertheless, they provided useful comments for the kick-off meeting of this study in May 2021 (see acknowledgement).

### 2.2. Study Design and Parameters

In the present retrospective cohort study, the primary endpoint was to clarify the associated parameters for postoperative critical complications or mortality, and the secondary endpoint was to clarify the associated parameters with intraoperative surgical records, such as blood loss, postoperative course as hospital stay period, and HCC-related patient outcomes during the short period of 3 years after CB in this series. The selection of the present subjects was based on the tumor location determined through preoperative diagnostic imaging, representing the anatomical information of HCC and liver functional reserve, performed at each institution. The KSGLS surveillance data comprised 171 CB cases, of which 142 cases were enrolled because they fulfilled all of the inclusion criteria, as shown in Figure 1. The remaining 29 patients were excluded because of deficient surgical or preoperative imaging data or an undefined indication for CB with respect to recognition of the case report form data by the principal author. The patient demographics, such as age, sex, cause of chronic liver injury, viral status, and previous treatments were collected. The preoperative physical status was assessed using the American Society of Anesthesiologists Physical Status (ASAPS) classification system. The preoperative liver function was assessed using the conventional liver functional parameters by the blood test, indocyanine green retention for 15 min (ICG-R15), Child–Pugh classification, and the liver damage grade defined by the liver cancer study group of Japan [11]. The tumor-related data included serum alpha-fetoprotein (AFP) and protein induced by vitamin K absence II (PIVKA-II) levels and histological findings by the 6th edition of the *General Rules for the Clinical and Pathological Study of Primary Liver Cancer* 2015 [12]. The preoperative tumor imaging parameters were assessed using enhanced computed tomography (CT) or magnetic resonance imaging at each institute, including: (1) tumor size or number; (2) tumor location; (3) distances between the main HCC and the intrahepatic main vasculatures, such as the hilar Glissonean pedicle and the right HV (RHV) trunk based on the radiological and pathological findings (Figure 2a,b); (4) existence of a tumorous compression of the hilar Glissonean pedicle, the HV trunk or confluence, inferior vena cava (IVC), and surrounding organs (Figure 2c–e); and (5) estimated resection volume by preoperative CT volumetry. The morphological classification patterns involving the portal pedicle and hepatic veins were classified using Qiu’s classification system for centrally located liver tumors (CLLTs). Ref. [13] Type I was characterized by the close proximity of the tumor to or direct invasion of the portal veins; type II was characterized by the close proximity of the tumor to, or direct invasion of the hepatic veins; type III was characterized by tumors not close to either portal veins or hepatic veins; and type IV was characterized by the close proximity of the tumor to, or direct invasion of both portal and hepatic veins. The surgical data were assessed based on the preoperative HCC treatment, anesthetic record, operating time, inflow-occlusion procedure and occlusion time, intraoperative blood loss (iBL), red blood cell transfusion, intraoperative adverse events, and issues from the additional segment 1 (caudate lobe) resection. The histological findings of HCC and non-tumor livers were evaluated using the 6th edition of the *General Rules for the Clinical and Pathological Study of Primary Liver Cancer* 2015 [12]. The notation for hepatectomy was determined according to the “New World” terminology by Nagino et al. in 2021 [7].

The postoperative patient outcomes were assessed based on the hepatectomy-related morbidity, mortality, and duration of in-hospital stay. The grades of post-hepatectomy liver failure (PHLF) and bile leakage (PHBL) were based on the criteria of the International Study Group of Liver Surgery (ISGLS) [14,15]. Prolonged ascites were defined as any ascites requiring intensive diuretics or an invasive evacuation procedure. Postoperative follow-ups included assessments of the AFP and PIVKA-II levels, as well as ultrasonography, CT, or magnetic resonance imaging every three months. Regarding the definition of local or distant HCC recurrence, each center requested that each radiologist make a diagnosis. If tumor recurrence was found, an optimum treatment was selected for patients with preserved liver function, according to the treatment policy at each institute. The existence of HCC relapse, cause of death, tumor-relapse-free status, and overall survival (OS) up to three years after hepatectomy were assessed to evaluate the quality of CB in this series. 

The potential sources of bias are the retrospective consecutive data collections at various institutions, which might have various operative procedures. The operative indications were also slightly different; however, we confirmed the significant differences in our series during the kick-off meeting.

### 2.3. Statistical Analysis

The basic continuous data are expressed as median values and as the mean ± standard deviation (SD) for comparison between groups or survival periods for univariate tests. Categorical data were compared using the chi-square test or Scheffe’s multiple comparison test. The data for each group were compared using a one-way analysis of variance (ANOVA), which was subsequently tested using Student’s *t*-test. The correlation between continuous data was tested using Spearman’s rank correlation test, and the correlation coefficient (r) was determined. Receiver operating characteristic (ROC) curves were constructed using the sensitivity against 1-specificity at each value, and the index of accuracy was calculated by the area under the ROC curve (AUROC), in which a value close to 1.0 indicated a high diagnostic accuracy. Subsequently, the cut-off value for each examined parameter according to the endpoints was defined. A univariate analysis identified potentially predictive variables using a significance level of *p* < 0.05. The identified factors were subsequently entered into a multivariate logistic regression analysis and the predictive value of the risk ratio was calculated. Regarding the factors related to iBL (>1500 mL), as a risk parameter of clinicopathological or surgical parameters, and the associated 10 parameters for PHLF and BL by univariate analysis (*p* < 0.10) were selected. The results of the multivariate analysis by applying significant parameters with a *p*-value < 0.05. Survival was analyzed from the day of surgery to the most recent follow-up (>12 months). Recurrence after surgery was determined based on the imaging findings. The OS and relapse-free survival (RFS) were examined using Kaplan–Meier curves. The statistical significance was set at *p* < 0.05. All statistical analyses were performed using the IBM Statistical Package for the Social Sciences (SPSS) for Windows version 23 (IBM Corp., Armonk, NY, USA).

## 3. Results

### 3.1. Study Population and Selected Patient Characteristics

Table 1 summarizes the perioperative clinicopathological, surgical, and prognostic characteristics of all 142 selected HCC patients who underwent CB. Eighteen patients (13%) had a history of non-surgical HCC treatment. The median values of tumor markers, such as AFP and PIVKA-II, tended to increase. HCC, mainly located in segment 4, was dominant (75%) in this series and nine cases involved segment 1 (caudate lobe). Thirty-nine patients (27%) had multiple HCCs, with tumor sizes ranging from 4 to 14 cm on preoperative imaging. Combined organ resections were performed in 5% of the patients. Intermittently, total hepatic inflow-occlusion (Pringle’s maneuver) was performed in 91% of the cases, the median operating time was 7.9 h, and the median iBL was 852 mL. In 20% of cases, the surgical margin of the tumor edge was positive. The median duration of the hospital stay was 26 days.

### 3.2. Relationship between Variables and Postoperative Morbidities and Mortality

Table 2 shows the relationship between major postoperative morbidities and the continuous and categorical parameters. With respect to PHLF, increased preoperative C-reactive protein (CRP) levels were associated with PHLF, in which the hospital stay period was significantly longer in PHLF grade C than in other grades (*p* < 0.05). ASA-PS 4, a tumor involving segment 1, a tumor size greater than 10 cm, radiological macroscopic vascular involvement of the portal or HVs, compression of the RHV, Qiu’s type IV, IVC, or diaphragm, iBL, and injury of the remnant vessel trunk were all significantly associated with PHLF grade BC (*p* < 0.05). PHLF was significantly associated with other hepatectomy-associated complications, such as organ surgical site infection (SSI), prolonged ascites, and in-hospital mortality (*p* < 0.05). With respect to post-hepatectomy PHBL, a low albumin level was a correlated parameter, and patients with BL had a longer hospital stay than those without BL. Notably, ASA-PS 4, radiological macroscopic vascular involvement of the portal vein or HVs, Qiu’s type I, and remnant vessel injury were significantly associated with PHBL grade BC (*p* < 0.05). PHBL was significantly associated with other complications, such as organ SSI and prolonged ascites (*p* < 0.05), but not with mortality. In terms of total severe postoperative morbidities, including PHLF grade BC and PHBL-grade BC, a correlated parameter was a low preoperative albumin level, and the existence of these morbidities was significantly associated with a longer hospital stay period (*p* < 0.05). 

Table 3 shows the multivariate analysis between the significant parameters (*p* < 0.05) and postoperative PHLF, PHBL, or severe morbidities using the univariate analysis described above. To avoid confounding effects with other precisely related parameters regarding vascular involvement, Qiu’s classification parameters were not applied. An iBL of >1500 mL was an independent parameter associated with PHLF, and the existence of middle and left HV confluency tended to be associated, but not significantly (*p* = 0.05). With respect to PHBL, a low albumin level was an independently associated parameter (*p* < 0.05), and iBL over 1500 mL tended to be associated with BL, but not significantly (*p* = 0.099). With respect to total severe morbidities, by the step-down procedure of the multivariable regression analysis, segment 1 resection, increased tumor size >6 cm, compression of the middle and left HV confluence, and compression of the IVC were selected as significantly associated parameters (*p* < 0.05). Glissonean pedicle stenosis tended to be associated with total severe morbidities; however, this was not significant (*p* = 0.053). With respect to iBL over 1500 mL, low albumin level <4 g/dL, high CRP ≥ 0.2 mg/dL, and large tumor size were independent associated parameters by the multivariate analysis (Table 4).

Four patients died in hospital within 90 days (2.8%) and one patient died within 30 days. All of the patients were observed to show optimal preoperative liver function to tolerate CB. Two patients had HCC in segment 1 (caudate lobe; H4581 or H4581), and two had large HCCs exceeding 10 cm in size. All HCCs compressed major vessels or the diaphragm. Grade C PHLF accompanied by prolonged ascites, PHBL, or any other manifestation was observed in all patients. One patient died of liver failure caused by remnant RHV occlusion due to severe compression of the tumor and venous injury within 30 days of hepatectomy.

### 3.3. Postoperative Patient Outcomes for Three Years

The mean and median durations of hospital stay were 34 ± 28 days and 26 days [interquartile range; 8–176]. For the survival analysis, 138 patients except four in-hospital deaths, were examined, as shown in Figure 3. Postoperative tumor relapse of HCC was observed in 71 patients (51%) (Table 1); additionally, 66 patients received various anti-HCC treatments. The cancer RFS rate at three years was 51% (Figure 3a), and the mean survival period was 45.4 ± 4.6 months. A cancer-free status was observed in 67 patients. Death due to cancer was observed in 25 patients, and death due to other diseases was observed in four patients at three years after hepatectomy (Table 1). The HCC-specific OS rate at three years was 81%, and the mean survival period was 161 ± 7.3 months in this series (Figure 3b).

## 4. Discussion

As major hepatectomy is limited by tumor location and liver functional reserve, the remnant liver must be retained even in cases of large HCCs centrally located in Couinaud’s segments 4, 5, and 8 (alias, mesohepatectomy), although the number of operative indications is limited [3,16]. HCC with capsule formation usually grows with an increased inflow and compresses the liver parenchyma, surrounding hepatic vasculature, or organs. In such cases, it is difficult to dissect the compressive part during transection between the vasculature and the surgical margin. In the case of single nodular type HCC, surgical margin positivity (0 mm) is usually allowed as a curative resection [17]. In comparison with extended hemi-hepatectomy or trisectionectomy with a severe risk of a smaller future remnant liver volume ratio (FLR) in comparison with CB, even by applying surgical/radiological manipulation to increase the future remnant liver volume, CB is considered as an alternative hepatectomy by obtaining a sufficient remnant liver volume. A comparison between CB and trisectionectomy may not be possible because of the background liver status and tumor condition during each hepatectomy. Hence, in the case of a large HCC adjacent to the main hepatic vasculature, it is occasionally difficult to achieve complete hemostasis along the exposed Glisson pedicle or hepatic venous trunk due to tumor compression. Severe injury to these main vasculatures leads to fatal outcomes, such as PHLF [18,19,20]. To our knowledge, the usefulness, limitations, and detailed risk factors of CB alone have not been fully elucidated because of the limited number of patients at each institute. To analyze this in a precise manner, we considered that multi-institutional collection by a retrospectively recent cohort is necessary even in a single-arm setting. We also examined cases since 2005, when the preoperative liver functional indication, hemostatic or dissected devices, and procedures in our district were well-established [9,21,22]. Post-hepatectomy hemorrhage (PHH) is an important liver-specific complication after major hepatectomy, according to the ISGLS definition. However, after checking the data of 142 patients, complications, such as PHH, were not observed, and no radiological or surgical interventions for PHH were observed. Eventually, postoperative blood transfusion for decreased hemoglobin levels was not remarkably performed in this series. 

The first important endpoint factor of this study design was to clarify the parameters related to major post-CB complications, such as PHLF, PHBL, and other morbidities requiring any pharmaceutical or interventional treatments. In the recent reports of several patient cohorts, the summarized Clavien–Dindo classification was evaluated as a risk outcome; however, post-hepatectomy morbidities varied by any associated cause [23,24]. First, the background liver injury level was supposed to be the most significant parameter of post-hepatectomy morbidities. As patients undergoing CB were selected based on these issues, portal hypertension, uncontrollable ascites, and huge splenomegaly were not observed in this series. Furthermore, based on histological findings, inflammatory or fibrotic injury levels were not significantly associated with morbidities in this analysis. Liver function parameters seemed to be more sensitive in predicting post-hepatectomy morbidities. The total severe morbidity indicated in the present study was equivalent to that in Clavien–Dindo grade III or higher. At this stage, the preoperative indications for hepatectomy and its extension were almost standardized in Japan; additionally, remarkable differences in ICG-R15, total bilirubin levels, and liver damage grade among institutes also seemed to be not observed in our 17 KSGLS institutes. Thus, the operative indications for liver function and the permitted resected liver volume were similar in the current series, and institutional indications or operative procedures have been well recognized in this district group by long-term investigations over 30 years. The correlations between the preoperative liver function parameters and morbidities were not significant in this study. However, once severe morbidities occurred, the in-hospital stay period was observed to be significantly longer. With respect to these severe morbidities, poor ASA-PS, segment 1 resection for expanded tumor involvement, and the compression of major trunk vasculatures were closely associated with higher grades of postoperative PHLF or PHBL. In the limited space under the subcostal and subphrenic areas, we verified that the difficulty of CB for occupied large HCCs, particularly those exceeding 10 cm, might be reflected by the iBL, longer operating time, or longer transecting time due to the difficulty of hemostasis at the transected area, exposure of major vessels and their repair. As a severe grade of PHLF, the most important lethal morbidity after hepatectomy was significantly associated with other morbidities, and the occurrence of each morbidity should be carefully considered at the time of preoperative assessments. In the present series, mortality was not observed to be higher than that in other reports of CB or major hepatectomy in patients with HCC [2,3,5]. This is the reason why careful management should be paid close attention to in each expert institute of our groups. In a small number of fatal cases, the common factors were male sex, tumor compression of major intra-or extra-hepatic vasculatures, and a surgical margin close to 0 mm, which eventually resulted in the death of PHLF grade C patients accompanied by other complications. One mortality case was the reason for starting the present cohort study, indicating in Figure 3 that trisectionectomy could not be estimated because of the small remnant liver volume of 30% of the whole liver. Considering such a fatal case, an up-front operative strategy should be avoided, and other non-operative treatments aimed at preventing tumor progression by arterial embolization or novel chemotherapy should be employed [8]. At this stage, the effectiveness or conversion rate of these methods for rapidly progressive HCC has not been fully recognized. Large resected volumes in hepatectomy and hepatic resection close to the IVC are critical anatomical points, even for experienced operators. Another source of concern was injury to the remnant major vasculatures in the case of CB for a larger HCC because any vascular issues indicated the possibility of lethal postoperative outcomes. It was difficult to achieve sufficient repair of the massively injured HV or its vascular reconstruction once the scheduled CB was set. Usually, the remnant liver functional volume can be secured in the case of CB compared to extended hemi-hepatectomy or trisectionectomy. However, if either of the liver sections (posterior section or the left lateral section) was impaired and limited, we must be aware of the fact that the remnant liver volume is too small to maintain sufficient functions to support life [25]. Although we hypothesized at the time of the study design that severe compression by a large HCC located centrally was related to lethal complications after CB, in the design of this study, we could not expect vascular compression to be associated with such a complication. As a result, iBL >1500 mL was the most associated risk factor for PHLF, which might be influenced by tumor compression of the middle and left HV confluence, but not the RHV trunk or root in the present study. Prior encircling of the RHV root is necessary; however, the RHV root is supposed to be a critical point of massive bleeding in cases of severe tumor compression [26]. Based on the present results, this anatomical point must be monitored for fatal massive bleeding and is related to hepatic function-related complications. 

In a complicated hepatectomy procedure, such as CB, the transected plane and exposed hepatic hilum are wide, and the possibility of increased blood loss or bile injury is considered to be high [14,27]. In the present series, we confirmed that most CB procedures precisely accomplished anatomical CB at each expert institute. In this situation, a lower albumin level is associated with BL, which may be possible because of fragile wound healing in the bile duct stump or a compressed hilar Glissonean pedicle. Under the kick-off meeting of our study group, the use of an energy device might not be applied adjacent to the large Glissonean pedicles around the hepatic hilum. As expected, the operative risk factors for CB were: (1) HCC involvement in H4581′, H4581, H4581-IVC, or more, (2) a large HCC, or (3) HCC compression of the confluence of the main HVs or IVC. In particular, HCC >10 cm in size with severe compression of the surrounding vasculature is considered lethal. We also stressed that, centrally located, a large HCC extending to segment 1 accompanied by tumor involvement adjacent to the HV confluence or IVC seems to be avoided by aggressive up-front CB, and other treatment modalities for HCC would be selected as a first-line strategy. In the present study, we originally evaluated the measurement of distances or involvement patterns between HCC and the main vasculature, such as the hilar Glissonean pedicles and hepatic veins or inferior vena cava. Qui et al. reported a simple classification regarding the involvement of HCC in major vessels, which was associated with postoperative outcomes [13]. This classification was associated with PHLF and PHBL in the univariate analysis. Because this classification is simple, it is easy to define the operative risk of vascular injury. We applied other original detailed parameters for multivariate analysis by considering statistical confounding factors. In cases of compressive HCC located in the central liver, various risk points may be considered. Vascular compression of HCC is thought to be possible to dissect due to morphological characteristics of the tumor capsule formation in comparison to adenocarcinomas. Surgeons may have challenged CB for compressive HCC in the present study. However, eight patients underwent combined vascular resection, which included the hilar hepatic duct in five patients undergoing hepaticojejunostomy, partial resection of the RHV in one patient, partial resection of the IVC in one patient, and graft replacement between the RHV and IVC in one patient. The last case was fatal because of liver congestion. Thus, the aim of the present study was to clarify and predict the parameters of operative risks for lethal morbidities.

With respect to PHLF, iBL >1500 mL can be defined as a predictive variable of outcomes [1,28,29]. As expected, a tumor size >10 cm and tumor compression adjacent to the roots of the exposed HV trunk, IVC, or hilar Glissonean pedicle were significant risk factors. We also expected that tumor involvement in segment 1, the caudate lobe, was a major risk factor for IVC and short hepatic venous injuries [29]. A large HCC located in segment 8 or 7 often severely compresses the right sub-diaphragmatic space. Additionally, it is difficult to handle and rotate the right liver because of the fixation between the subcostal arches. Nevertheless, by combining additional thoracotomy, the difficulty of surgery might not be improved in the case of large HCC >10 cm [30]. An iBL >1500 mL was significantly related to poor post-hepatectomy outcomes, which might also be related to tumor recurrence. In patients with HCC, low albumin levels often occur because of background chronic liver injury, which is thought to reflect liver parenchymal fragility. A tumor size >10 cm and the risk of larger iBL were significantly increased in the present study, as well as in previous reports [3,16,29,30,31], which were clarified in CB in the present study.

To confirm the short-term patient prognosis, the three-year-survival after CB was indicated. Although the recurrence rate of HCC was 50% at three years, influenced by the presence of large or multiple HCCs, the OS for three years was met, which was better than other results [3,10], owing to the careful follow-up management and decision of optimal anti-HCC treatments at these expert institutes. This result might indicate good survival benefits as an alternative to right or left trisectionectomy. A longer prognosis must be expected as basic data before the recent development of systemic chemotherapy or other treatments for advanced HCC. By considering the survival outcomes after CB in centrally located HCC and the relationship with the oncological point of view as the surgical margin, we focused on clarifying the operative risk first, and poor prognostic factors were examined in 3- or 5-year survival in the next step. At this point, HCC-related pathological factors or tumor markers might be significantly associated with the tumor recurrence rate or 3-year survival, but not the operative risk parameters clarified in the present study (data not indicated and described in this manuscript according to the primary aim or endpoint of the study).

### Limitations

This study has a few limitations, as follows: (1) The present study was a retrospective observational study that collected patient data. The study included various institutional indications and operative procedures; however, the study protocol was fully discussed in the first step of our study group. The clinical status of HCC may differ between countries or races. Therefore, it is necessary to consider that the present study is limited to a district of Japanese (Asian) populations with a relatively small sample size for readers, although the attending 17 institutes specialized in liver surgery. A disparity of operative procedures or devices for hepatic parenchymal transection was considered, which might have influenced the surgical records and patient outcomes. As described above, the number of CBs was very limited or slightly different between institutes; however, it was insufficient to accomplish a statistically precise analysis using the given data, even with 15 years at a single or several institutions. Therefore, the data were comprehensively collected from 17 institutes during our kick-off meeting of the KSGLS study group. As we have annually convened our study group conferences for over 30 years, the quality of liver surgery is considered to be reliable. Compared with a national database with limited analytical parameters, further detailed parameters can be obtained by a regional but reliable multi-institutional study. (2) Numerous heterogeneous analytical parameters were observed. In the case of CB, more factors associated with blood loss, difficulty of operation, morbidities, and prognosis were considered, making it difficult to select specific parameters. However, novel and essential risk factors for CB can be identified in several candidates. With the help of several web meetings, the obtained data were well discussed by the study group, and we concluded that the present data were reasonable. (3) Surgical solutions to the problems clarified by our results have not been clearly indicated. However, the efficacy of novel neoadjuvant molecular targeting or immune checkpoint inhibitor chemotherapy, including combined multimodal interventional chemoembolization before CB, has not been established. Novel adjuvant systemic or local regional therapy may reduce the tumor volume. Conversion chemotherapy for advanced HCC has recently become extremely promising [32,33,34]. This adjuvant modality was not used in this study. A large HCC that is observed to rapidly progress and compress its surroundings, in particular, should be treated urgently with novel chemoembolization. Hence, the operative indications must be considered by checking responses. To date, this treatment strategy has not been fully elucidated in the next step.

## 5. Conclusions

In conclusion, this multi-institutional retrospective cohort analysis revealed operative risk factors in 142 HCC patients who underwent CB (H458). The involvement of HCC in segment 1 (H4581′ or H4581), tumor size >10 cm, and tumor compression to the main HV vasculatures were independent risk factors for PHLF after CB and were associated with iBL >1500 mL. The 3-year survival rate after the CB treatment was favorable. Although CB has been recognized as an alternative or technically difficult hepatectomy for centrally located HCC, when compared to extended hemi-hepatectomy, this anatomical operative option must be important and curable under precise preoperative simulations, elaborate or careful transection techniques, and perioperative management at an expert institute. These results suggest exploring the borderline limits of challenging up-front CB, which may provide useful information to determine other optimal non-operative novel treatment modalities.

## Figures and Tables

**Figure 1 cancers-15-01740-f001:**
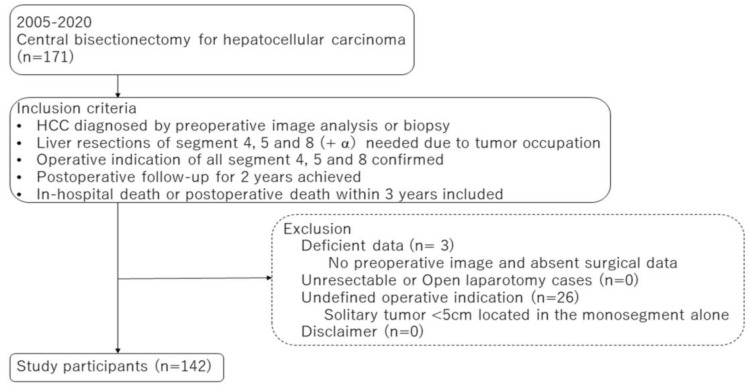
Inclusion criteria for collecting the cohort data of HCC patients undergoing CB (H458) from 17 expert institutions of the Kyushu Study Group of Liver Surgery as a single-arm study. CB, central bisectionectomy; HCC, hepatocellular carcinoma.

**Figure 2 cancers-15-01740-f002:**
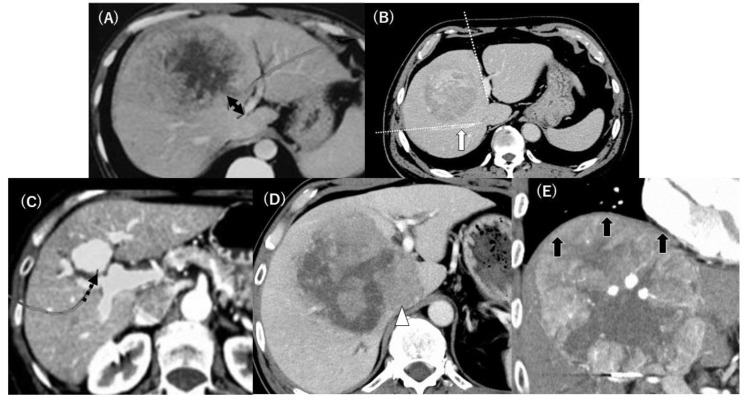
Representative HCC sample of the computed tomography (CT) images for CB and investigated items in the case report form for each institute: (**A**) the distance or existence of tumor compression to the hilar Glissonean pedicle (black double arrow), (**B**) the adequate transected planes required in CB (dotted white lines) and the right hepatic vein trunk and the distance or existence of compression by the tumor (white arrow), (**C**) the existence of tumor involvement in the portal or bile duct at the hepatic hilum (dotted black arrow), (**D**) tumor involvement in segment 1 (caudate lobe) except segment 4,5, and 8 or compression of IVC (white arrow head), (**E**) tumor compression of the diaphragm (black arrows). CB, central bisectionectomy; HCC, hepatocellular carcinoma.

**Figure 3 cancers-15-01740-f003:**
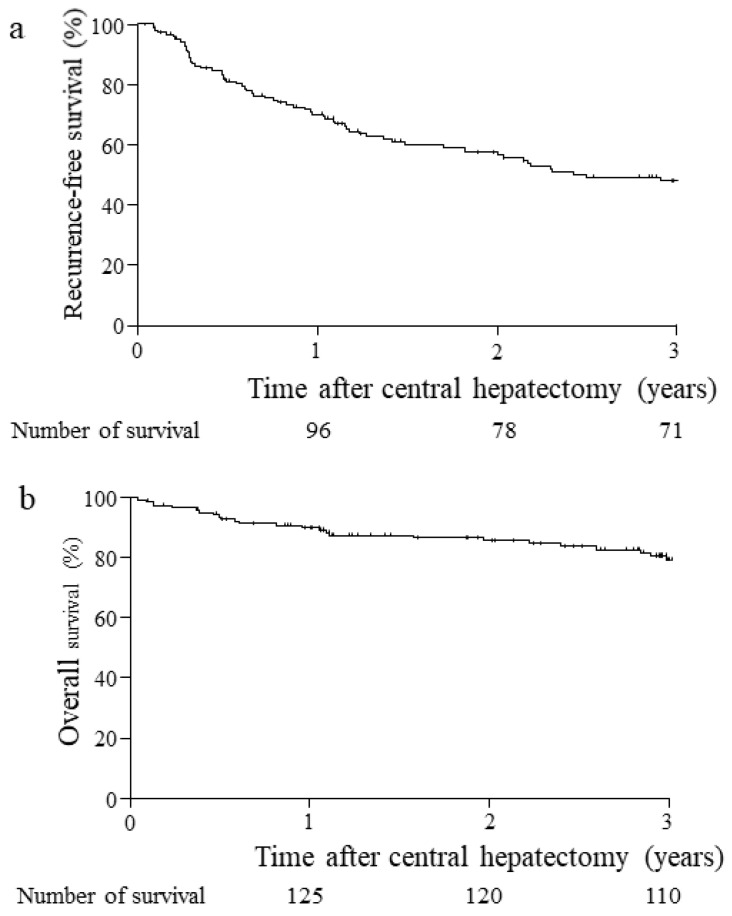
Number of patients who survived are indicated. Tumor recurrence-free survival (**a**) and overall survival (**b**) within three years in 138 HCC patients who underwent CB, except four cases of in-hospital deaths after CB. CB, central bisectionectomy; HCC, hepatocellular carcinoma.

**Table 1 cancers-15-01740-t001:** Clinical, laboratory, radiological, and pathological characteristics of all subjects who underwent central bisectionectomy (H458).

Variables (*n* = 142)	*n*, (%), Median Value [Minimum–Maximum]
Age (years)	73 [36–87] *
Sex (Male/Female)	118 (83)/24 (17)
ASA-PS (*n* = 137)	
1/2/3	30 (22)/93 (68)/14 (10)
Background liver status	
Chronic liver injury, non/chronic hepatitis/cirrhosis	39/103 (73)
non-BC (NAFLD 22, PBC 1, other 7)	72 (51)
B	21 (15)
C	49 (34) ^#^
Portal hypertension, yes	4 (3)
Ascites, yes	2 (1) (controllable, mild)
Splenomegaly, no/mild/moderate or more	122 (85)/19 (14)/1 (1)
Preoperative liver functions	
Child–Pugh classification	A 138/B 4 (3)
Liver damage grade	A 121/B 14 (10)
ICGR15 (%)	11.2 [1.3–25]
Total bilirubin (mg/dL)	0.7 [0.3–2.6]
Albumin (g/dL)	4.1 [2.3–4.9]
Prothrombin activity (%)	95 [44–139]
Platelet counts (× 104/mm^3^)	18.2 [9.8–77]
C-reactive protein (mg/dL)	0.19 [0.02–10.1]
Alfa-feto protein (ng/mL)	13.3 [1.1–383,541]
PIVKA-II (mAU/mL)	323 [9–269,391]
Tumor location diagnosed before operation	
Main location, S8/S4/S5	25 (18)/107 (75)/10 (7)
S1 involved, yes	9 (6)
Number of HCC	1.2 [range 1–6] *
1/2/3/4/5/6	103 (73)/24 (17)/8 (6)/2 (1)/2 (1)/3 (2)
Size	6.0 [range 4–14] *
0–4.9/5–9.9/over 10 cm	1 (1)/10 (6)/160 (93)
Macrovascular involvement no/yes §	130 (92)/12 (8)
Portal vein no/yes	136 (96)/6 (4)
Hepatic venous no/yes	134 (94)/8 (6)
Distance to major vessels	
Hilar Glisson, >10/10- >0/0 mm	106 (75)/24 (17)/11 (8); Stenosis, 7
MHV, >10/10- >0/0 mm	96 (68)/13 (9)/32 (23)
RHV, >10/10- >0/0 mm	115 (82)/12 (8)/14 (10)
IVC >10/10- >0/0 mm	131 (93)/7 (5)/3 (2)
Right diaphragm involvement, yes	26 (19)
Qiu’s classification system of centrally located HCC ¶	
Type I/II/III/IV	7 (5)/49 (35)/75 (53)/5 (3)/6 (4)
Preoperative cancer treatment	
TAE/chemotherapy/brachytherapy	15 (11)/1 (1)/2 (2)
Surgical records	
Additional resection except H4581, yes	8 (6)
S1 combined resection, yes	8 (6)
Estimated resection volume (cm^3^)	380 [28–1243]
Expose hilar plate Glisson sheath, yes	93 (66)
root of MHV/entire RHV/front of IVC	125 (87)/100 (71)/51 (36)
Hepatic inflow occlusion (Pringle), no/yes	13/129 (91)
Time (min)	115 [15–322]
Taping of main HV root, yes	18 (13)
CVP before transection (mmHg) (*n* = 88)	6.0 [1–11]
Operating time (min)	475 [282–1807]
Intraoperative blood loss factors (iBL)	
Intraoperative blood loss (mL)	852 [42–23,336]
Red cell transfusion, no/yes	88/54 (38)
Injury of preserved main vessels, yes	15 (11)
Histology	
Diagnosis (HCC, combined type HCC)	136/6 (4)
Occupied location	
Main location, S8/S4/S5	25 (18)/107 (75)/10 (7)
S1 involved, yes	9 (6)
Histological tumor findings	
F (fibrotic grade) ^‡^ 0/1/2/3/4 (*n* = 98)	15 (11)/50 (37)/28 (21)/25 (18)/18 (13)
Normal liver/chronic hepatitis/cirrhosis	1 (11)5/109 (76)/18 (13)
A (necro-inflammatory grade) ^‡^ 0/1/2/3	15 (15)/56 (57)/24 (25)/3 (3)
Surgical margin (mm)	4.0 [0–26]
0/0.1–5/>5 mm	29/58/55
Postoperative outcomes	
Complications	
Liver failure (PHLF), no/A/B/C **	85 (60)/39/12/6
Bile leakage (PHBL), no/A/B/C **	87 (61)/28/26/1
Prolonged ascites >7 days, yes	18 (13)
Post-hepatectomy hemorrhage (PHH)	
Radiological or surgical intervention, yes	nil
Organ space SSI, yes	21 (15)
Others	32
Mortality, yes	4 (3)
Within 30 days, yes	1 (0.5)
Hospital stay (days)	26 [8–176]
Prognosis within 3 years	
1st cancer recurrence, yes	71 (51); liver 60, extra-liver 10, lung 1
Relapse-free survival period (days)	1362 ± 139 (mean ± standard deviation)
rate (1-, 2-, 3-year)	70%, 56%, 48%
Prognosis	
Cancer-free survival	109
Dead, in-hospital/HCC/other disease	4/25/4
Overall survival period (days)	4838 ± 216 (mean ± standard deviation)
rate (1-, 2-, 3-year)	90%, 86%, 81%

* All continuous data are indicated by the median value because of non-normal distributions of most parameters. ** ISGLS classification grade [13,14], # This group included both hepatitis B and C infection in one. ¶ A classification system of centrally located liver tumors C (CLLTs) [13]. Abbreviations: HCC, hepatocellular carcinoma; ICGR15, indocyanine green retention test; PIVKA-II, protein induced by vitamin K antagonist or agonist; ASA–PS, American Society of Anesthesiologists Physical Status; CVP, central venous pressure; NAFLD, non-alcoholic fatty liver disease; PBC, primary biliary cholangitis; S, liver segment; HV, hepatic vein; MHV, middle hepatic vein; RHV, right hepatic vein; IVC, inferior vena cava; PHLF, liver failure; PHBL, bile leakage; SSI, surgical site infection. Histological fibrotic index ‡ [12] § Tumor thrombus or invasion of the main portal tract or its 2nd branch was detected by preoperative imaging findings.

**Table 2 cancers-15-01740-t002:** Relationship between variables and postoperative major morbidities in 142 patients undergoing CB (H458).

Variables	PHLF				PHBL				Total Morbidity #	
	None	A	B	C	None	A	B	C	No	Yes
Demographics										
ASA-PS										
1	0	5	0	1	5	0	1	0	6	5
2	3	16	3	2	5	15	4	0	15	9
3	68	16	8	1	65	12	15	1	56	37
4	9	2	1	2 **	8	1	5	0 **	5	9
Liver functions										
Liver damage grade										
A	78	30	9	4	73	25	22	1	74	8
B	6	4	3	1	8	3	3	0	8	6
ICGR15 (%)	12.2 (6.6)	12.6 (6.6)	24.5 (16.9)	9.7 (3.8)	13.6 (8.3)	14.9 (10.6)	11.1 (6.8)	4.1	14.0 (8.2)	12.5 (9.2)
Bilirubin (mg/dL)	0.80 (0.3)	0.83 (0.4)	0.84 (0.4)	0.77 (0.3)	0.80 (0.3)	0.81 (0.3)	0.85 (0.5)	0.6	0.79 (0.3)	0.83 (0.4)
Albumin (g/dL)	4.0 (0.4)	4.2 (0.4)	3.9 (0.5)	3.8 (0.9)	4.0 (0.4)	4.3 (0.3)	3.9 (0.6) *	3.7	4.1 (0.4)	3.9 (0.5) **
PT (%)	95 (17)	94 (11)	87 (13)	90 (10)	94 (15)	93 (10)	94 (19)	88	94 (15)	94 (15)
Platelet (/mm3)	19.6 (7.2)	20.4 (7.5)	24.3 (17.3)	20.9 (7.3)	20.1 (9.6)	20.9 (6.7)	19.8 (6.7)	29.2	20.7 (9.9)	19.7 (6.2)
CRP (mg/dL)	0.57 (1.5)	0.39 (0.4)	0.27 (0.3)	2.23 (2.8) *	0.57 (1.5)	0.35 (0.4)	0.8 (1.5)	0.55	0.50 (1.4)	0.65 (1.3)
Imaging of tumor										
Location (main)										
S8	13	11	1	0	15	10	0	0	20	5
S4	64	27	11	5	65	17	24	1	56	51 *
S5	8	1	0	1	7	1	2	0	6	4
S1 involvement (H4581), yes	5	1	2	2 *	8	1	1	0	3	7
Number of tumors										
1	61	30	9	3	62	22	18	1	57	46
2	15	5	1	3	14	5	5	0	16	8
3≤	9	4	2	0	11	1	3	0	9	6
Tumor size (mm)										
0–5	37	6	4	1	37	5	5	1	29	19
5–10	38	24	3	2	37	15	15	0	39	28
10<	10	9	5	3 **	13	8	6	0	14	13
Vascular invasion										
0	73	23	11	3	76	15	19	0	61	49
1	4	14	1	1	5	11	3	1	13	7
2 and 3	7	1	0	2 **	6	1	3	0 **	8	4
Portal vein										
0	78	27	12	3	80	19	21	0	68	52
1	5	10	0	1	6	7	2	1	11	5
2 and 3	2	2	0	2 **	1	2	3	0 **	3	3
Hepatic vein										
0	76	31	11	5	80	22	20	1	70	53
1	2	8	1	0	2	6	3	0	7	4
2 and 3	7	0	0	1	5	0	3	0	5	3
Glisson pedicle stenosis										
no	81	37	11	5	84	27	23	0	79	55
yes	4	2	1	1	3	1	3	1 **	3	5
Compression of LMHV root										
no	83	32	8	6	81	24	23	1	74	55
yes	2	7	4	0 **	6	4	3	0	8	5
RHV root										
no	80	36	10	3	77	27	24	1	76	53
yes	5	3	2	3 **	10	1	2	0	6	7
IVC										
no	84	38	10	5	84	26	26	1	79	58
yes	1	1	2	1*	3	2	0	0	3	2
Diaphragm										
no	75	29	9	3	74	21	20	1	68	48
yes	10	10	3	3*	13	7	6	0	14	12
Qiu’s CLLTs type										
I	4	2	1	0	3	1	2	1*	4	3
II	25	18	4	2	29	12	8	0	30	19
III	55	17	5	2	52	13	14	0	47	32
IV	1	2	2	2*	3	2	2	0	1	6
Treatment related factors										
Resected volume (cm^3^)	355 (253)	505 (223)	563 (395)	723 (485)	62.9 (32.3)	70.9 (36.3)	73.9 (28.3)	709	419 (293)	420 (275)
Segment 1 partial resection										
no	81	37	11	5	83	27	23	1	79	55
yes	4	2	1	1	4	1	3	0	3	5
Occlusion time (min)	110 (57)	87 (66)	124 (91)	116 (84)	374 (289)	546 (275)	440 (239)		99 (62)	114 (67)
Operating time (min)										
<480	45	22	5	2	49	14	11	0	42	32
≥480	40	17	7	4	38	14	15	1	40	28
Blood loss (mL)										
<1500	68	25	5	1	66	18	15	0	61	38
≥1500	17	14	7	5 **	21	10	11	1	21	22
Histological factors										
Fibrotic grade										
0	7	8	0	0	6	5	4	0	8	7
1	31	12	6	1	33	11	5	1	31	19
2	18	5	3	2	17	2	9	0	16	12
≥3	25	12	3	3	28	8	7	0	22	21
Necroinflammatory grade (*n* = 98)										
0	9	6	0	0	9	3	3	0	8	7
1	32	15	5	4	35	12	9	0	36	20
≥2	16	5	4	2	19	4	4	0	14	13
Background liver status										
normal	7	8	0	0	6	5	4	0	8	7
chronic hepatitis	64	28	11	6	67	21	20	1	64	45
cirrhosis	14	3	1	0	14	2	2	0	10	8
Postoperative outcomes										
Hospital stay (days)	30 (24)	35 (29)	39 (22)	66 (56) †	27 (14)	27 (30)	64 (37) ‡	74	23 (10)	49 (36) **
Complications										
Organ SSI										
no	76	31	11	3	85	27	9	0	-	-
yes	9	8	1	3 *	2	1	17	1 **		
Prolonged ascites										
no	79	35	7	3	78	25	21	0	-	-
yes	6	4	5	3 **	9	3	5	1 **		
Mortality										
no	85	39	12	2	86	28	23	1	-	-
yes	0	0	0	4 **	1	0	3	0		

† *p* < 0.05 vs. not significant, ‡ *p* < 0.01 vs. no & A * *p* < 0.05, ** *p* < 0.01. Categorical and continuous data were combined. In continuous data, the mean value was indicated and parenthesis showed the standard deviation (+/− SD). The grades of PHLF and BL were based on the criteria of the International Study Group of Liver Surgery (ISGLS) [13,14]. # The vascular involvement level or histological background factors were referred to the *General Rules for the Clinical and Pathological Study of Primary Liver Cancer* by the Liver Cancer Study Group of Japan, 6th edition [12]. The Yes group was the number of total complications consisting of PHLF grade BC, PHBL grade BC, prolonged ascites over 7 days, organ space SSI, or other severe complications that required aggressive drug and blood access treatments or surgical, radiological, and endoscopic treatments (equivalent to Clavien–Dindo classification grade IIIa or more).

**Table 3 cancers-15-01740-t003:** Multivariate logistic regression analyses between clinical and surgical parameters and major postoperative morbidity in CB.

Variables	Significance	RR	95% Confidence Interval	
	(*p* Value)		Lower Limit	Upper Limit
PHLF				
CRP ≥ 0.2 mg/dL	0.379	1.43	0.644	3.171
Tumor size ≥ 10 cm	0.556	1.473	0.405	5.356
Segment 1 resection, yes	0.254	2.347	0.541	10.18
Vascular involvement of tumor, yes	0.475	0.629	0.177	2.242
Compression of middle & left HV confluence, yes	0.05	6.818	0.995	46.93
Compression of RHV, yes	0.928	1.073	0.236	4.886
Compression of IVC, yes	0.472	0.338	0.018	6.502
iBL ≥ 1500 mL	0.025	2.791	1.14	6.826
PHBL				
Albumin < 4 g/dL	0.009	2.994	1.31	6.849
Tumor size ≥ 10 cm	0.255	1.809	0.651	5.028
Glissonean pedicle stenosis, yes	0.128	6.976	0.572	85.05
Segment 1 resection, yes	0.092	2.662	0.852	8.319
Inflow hepatic occlusion ≥ 120 min	0.914	1.045	0.472	2.314
iBL ≥ 1500 mL	0.099	2.072	0.871	4.93
Total severe morbidities #				
Step 1				
Total bilirubin ≥ 0.8 mg/dL	0.776	1.208	0.329	4.432
Albumin < 4 g/dL	0.291	0.498	0.136	1.816
CRP ≥ 0.2 mg/dL	0.159	2.795	0.669	11.674
Segment 1 resection, yes	0.057	10.2	0.933	111.43
Tumor size ≥ 6 cm	0.071	3.562	0.897	14.14
Vascular involvement of tumor, yes	0.799	1.345	0.138	13.15
Glissonean pedicle stenosis, yes	0.205	7.149	0.341	150.04
Compression of middle & left HV confluence, yes	0.02	33.59	1.76	642.92
Compression of RHV root, yes	0.255	3.602	0.397	32.7
Compression of IVC, yes	0.037	0.008	0	0.755
Inflow hepatic occlusion ≥ 120 min	0.972	1.023	0.292	3.589
Operation time ≥ 480 min	0.164	0.367	0.089	1.506
iBL ≥ 750 mL	0.132	2.968	0.72	12.23
Red cell blood transfusion, yes	0.847	0.853	0.169	4.299
Step 6 ***				
Segment 1 resection, yes	0.037	5.67	1.11	29.056
Tumor size ≥ 6 cm	0.013	3.749	1.33	10.589
Glissonean pedicle stenosis, yes	0.053	6.505	0.976	43.331
Compression of middle & left HV confluence, yes	0.032	8.948	1.21	66.002
Compression of IVC, yes	0.049	19.61	1.01	333.33

# See Table 2. *; Calculated by the step-down procedure of logistic multiple regression analysis. SE, standard; RR, risk ratio; other abbreviations, see Table 1.

**Table 4 cancers-15-01740-t004:** Relationship between perioperative continuous or categorical parameters, and increased blood loss (iBL) over 1500 mL in CB through multivariate logistic regression analyses.

	iBL ≥ 1500 mL			
	Significance	RR	95% Confidence Interval	
Variables	(*p* Value)		Lower Limit	Upper Limit
Step (1)				
Albumin <4 g/dL	0.18	5.139	1.328	19.895
PT < 90%	0.559	0.691	0.2	2.392
CRP ≥ 0.2 mg/dL	0.028	4.901	1.188	20.213
Estimated liver volume ≥ 500 cm^3^ *	0.065	4.745	0.907	24.824
Tumor size ≥ 6.2 cm	0.043	6.25	1.058	37.037
≥10 cm	0.04	10.47	1.114	98.36
Compression of RHV root, yes	0.942	1.137	0.036	36.258
Diaphragm compression by HCC, yes	0.054	0.105	0.011	1.042
Operating time > 480 min, yes	0.169	3.414	0.594	19.618
Resection of segment 1, yes	0.051	9.598	0.988	93.254
Step (11) **				
Albumin < 4 g/dL	0.017	3.795	1.267	11.374
CRP ≥ 0.2 mg/dL	0.03	3.74	1.137	12.304
Tumor size ≥ 6.2 cm	0.025	6.25	1.309	29.412
≥10 cm	0.002	7.178	2.358	37.346

* Preoperative predicted volume. **; calculated by the step-down procedure of the logistic multiple regression analysis. SE, standard; RR, risk ratio; other abbreviations, see Table 1.

## Data Availability

The datasets generated and/or analyzed during the current study are available from the corresponding author upon reasonable request.

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
