# Peer review of "Risk Factors of Complications from Central Bisectionectomy (H458) for Hepatocellular Carcinoma: A Multi-Institutional Single-Arm Analysis"

_cancers, 2023, doi:10.3390/cancers15061740_

Round 1

Reviewer 1 Report (Previous Reviewer 2)

The authors addressed all the queries and issues we raised.

We recommend Acceptance.

Reviewer 2 Report (Previous Reviewer 3)

The authors responded to all comments. No further comments are to be addressed.

This manuscript is a resubmission of an earlier submission. The following is a list of the peer review reports and author responses from that submission.

Round 1

Reviewer 1 Report

Dear editor,

With great interest, the article entitled "Specific Risk Factors of Severe Complications in Central Bisectionectomy (H458) for Hepatocellular Carcinoma: A Multi-institutional Single-arm Analysis" is reviewed. The report was aimed to evaluate the factors, which play a role in the prediction of postoperative morbidity and mortality among patients with hepatocellular carcinoma (HCC) undergoing central-bisectionectomy. The authors have revealed that tumors larger than 6cm, macrovascular involvement or compression and introperative blood loss >1500ml are associated with higher rate of major postoperative complications, in these patients.

Central-bisectionectomy is considered as a surgical technique for resection of centrally located liver tumors. Although its parenchyma-preserving nature might be of grave importance for patients with underlying liver diseases, including HCC cases, defining the predictive and risk factors of this procedure should be not be limited only to HCC patients.  It is clear that the authors have aimed to assess the outcomes in a homogenous group of patients, but the main question as well as need of the study are not justified. Recently, several western studies with larger sample size have evaluated the short- and long-term outcomes of  the mesohepatectomy as well as their predictors, questioning the novelty of the current study. In addition, oncological outcomes of mesohepatectomy in HCC are also not focus of the study. Thus, it is not clear, whether they aimed to emphasize the surgical or oncological aspect  of mesohepatectomy in HCC. If the authors aim to discuss the surgical characteristics of the mesohepatectomy, I would suggest them to include all indication in their analyses, and rather carry on subgroup analyses, for different indications. So, they benefit a larger sample size and stronger conclusions. In case of evaluating the oncological outcomes, further data is needed, in means of resection margin, median follow-upm site of recurrence, and etc. Also, there are some major concerns about the manuscript, which demands the authors' consideration.

1.     The main problem of the manuscript is the data presentation. Particularly, the presentation of uni- and multivariate analyses make the interpretation and digestion of the data almost impossible. Especially in Table 2, please consider to provide each variable or subvariable in separate lines.

2.     The authos have not mentioned or discussed “posthepatectomy hemorrhage (PHH)”, at any part of the manuscript. This is also one of the important liver-specific complications, introduced by ISGLS. Why the rate and severity of PHH were not included in the data?

3.     Please specify the information regarding the postoperative follow-up. How were the intervals for follow-up and what kind of examinations or work-ups were carried out.

4.     Regarding the patients with HCC, underlying liver diseases, including cirrhosis, hepatitis, and etc., are also important factors, which have impact on perioperative outcomes of the patients. More detailed presentation of these data and subsequent consideration in analyses is needed.

5.     Please provide the tumor location as well as type of mesohepatectomy based on the classification provided by Qiu et al. (Surg Oncol. Qui et al., 2016 Dec;25(4):441-447. doi: 10.1016/j.suronc.2016.03.001.)

6.     Many variables and their categorization are not mentioned in methods and materials. For instance, the references for categorization of tumor distance to vascular structures and their involvement, should be provided.

7.     It should also be reported, how many vascular resections were carried out, if any. Besides, tumor compression and tumor ifiltration to major vessels are two completely different entities, which cannot taken into account similarly, since intraoperative approaches to these conditions are totally different. 

8.     Single patient had a intraoperative bloods loss of 23L and duration of OP was 30 hours. Please provide further details regarding this case.

9.     Please use the standard and well-known abbreviation and terminology for description of the ouctomes including PHBL and PHLF. As shown in the figures, the outcomes of Kaplan-Meier analysis should be reported in the Results section or a table in detail.

Author Response

February 28, 2023

 Dear Editor-in-chief, and editorial office staff

Cancers,

cancers-2229392R1; Manuscript Title: Risk Factors of Complications in Central Bisectionectomy (H458) for Hepatocellular Carcinoma: A Multi-institutional Single-arm Analysis (the title was revised according to reviewer’s comment)

Principle author: Atsushi Nanashima

 Dear editor-in-chief,

Please find enclosed the above resubmitted manuscript to Cancers as a full article.  

We are so sorry to know that the present paper has been rejected under your web site comments. According to reviewer’s comments, however, possibility of resubmission may remained and two of three reviewers might potentially accepted our manuscript. One reviewer who denied reviewing also showed the forward opinions in the comment. Thus we have attempted resubmission according to these comments which was revised by point-by-point response by yellow highlighted as the editor mentioned.

After the initial submission, the journal’s office pointed out that the personal information as case series report was shown in Table 5 and Figure 3. These were indicated as representative cases of mortality at first. The office required the acceptance with assignment of family’s name and, however, we cannot contact all four cases actually. Therefore, we would like to request deletion of these data in re-submission manuscript. The authors also understand that they will automatically transfer the copyright to the publisher. We look forward to publication of our revised manuscript. The authors also understand that should be submitted material been accepted for publication in the journal, they will automatically transfer the copyright to the publisher.

I hope that the reviewing process finds the resubmitted manuscript acceptable for publication in the journal.

 Sincerely,

Professor Atsushi Nanashima, M.D., PhD., and FACS.

Division of Hepato-biliary-pancreas surgery, Department of Surgery, University of Miyazaki Faculty of Medicine, Kiyotake 5200, Miyazaki, 889-1692, Japan.

E-mail; a_nanashima@med.miyazaki-u.ac.jpTelephone: +81-985-85-2905;Fax: +81-985-85-3780

 cancers-2229392R1; Manuscript Title: Risk Factors of Complications in Central Bisectionectomy (H458) for Hepatocellular Carcinoma: A Multi-institutional Single-arm Analysis (the title was revised according to reviewer’s comment)

Response to reviewer 1

We are so sorry to know that the reviewer 1 would not like to sign my review report and, however, very carefully checked our manuscript and gave us useful comments or suggestions. This can be revised. We suppose that resubmission would be reconsidered by the same reviewer’s sincere comment. Revised part was indicated by yellow highlighted accompanied with point-by-point responses as below.

  1. English language and style are fine/minor spell check required

Yes        Can be improved Must be improved. English language in the entire text was newly edited by Elsevier editing service (line 566-568).

1) As other reviewer suggested, the entire text were edited by the native scientists.

Does the introduction provide sufficient background and include all relevant references?

( )          ( )          (x)         ( )   This was revised according to the reviewer’s comments

Are all the cited references relevant to the research?

( )          (x)         ( )          ( )

Is the research design appropriate?

( )          (x)         ( )          ( )

Are the methods adequately described?

( )          ( )          (x)         ( )   This was revised according to the reviewer’s comments

Are the results clearly presented?

( )          ( )          (x)         ( )   This was revised according to the reviewer’s comments

Are the conclusions supported by the results?

( )          (x)         ( )          ( )

2) Central-bisectionectomy is considered as a surgical technique for resection of centrally located liver tumors. Although its parenchyma-preserving nature might be of grave importance for patients with underlying liver diseases, including HCC cases, defining the predictive and risk factors of this procedure should be not be limited only to HCC patients.

In addition, oncological outcomes of mesohepatectomy in HCC are also not focus of the study HCC. Thus, it is not clear, whether they aimed to emphasize the surgical or oncological aspect of mesohepatectomy in HCC. If the authors aim to discuss the surgical characteristics of the mesohepatectomy, I would suggest them to include all indication in their analyses, and rather carry on subgroup analyses, for different indications. So, they benefit a larger sample size and stronger conclusions. In case of evaluating the oncological outcomes, further data is needed, in means of resection margin, median follow-up site of recurrence, and etc. Also, there are some major concerns about the manuscript, which demands the authors' consideration.

  1. a) Regarding to surgical technique, other liver diseases would be included. However, cholangiocarcinoma was relatively rare, metastatic liver cancers tended to be limitedly resected and, therefore, in these expert 17 institutes, subject undergoing CB was not mostly observed. This comment was added in the Introduction (line 61-75).
  2. b) In this study, surgical aspects relating posthepatectomy outcomes during hospitalization was focused, but not oncological point of view. Therefore, related issues relating tumor relapse or overall survivals were not examined in this manuscript. This was described in Introduction (line 75-80).
  3. c) With respect to evaluation of oncological point of view, we examined unpresented data during 3 years. However, only tumor related pathological findings or tumor marker only related to the survivals but not significant with surgery related parameters in this series. This point was a bit described with future aspects in Discussion (line 484-491).

3) The main problem of the manuscript is the data presentation. Particularly, the presentation of uni- and multivariate analyses make the interpretation and digestion of the data almost impossible. Especially in Table 2, please consider to provide each variable or subvariable in separate lines.

We are sorry not presenting well in Table 2 or others because of various and lot of parameters. First of all, each variables and sub-variables were presented in separated lines (Table 2). Further, according to another reviewer’s comment, all data were combined in revision.

4) The authors have not mentioned or discussed “posthepatectomy hemorrhage (PHH)”, at any part of the manuscript. This is also one of the important liver-specific complications, introduced by ISGLS. Why the rate and severity of PHH were not included in the data?

We appreciate your instruction and suggestion. PHH is one of the important liver-specific complications after major hepatectomy. By checking data of 142patients, complication as PHH was not indicated and no radiological or re-operative intervention for PHH was not observed. Although other severe complications were reported from each institute, postoperative blood transfusion for decreased hemoglobin by PHH was not remarkable in each institute. Therefore, as we know, data of PHH was added in Table 1 but not analyzed as a subject in this series. This issue was described in Discussion (361-366).

5) Please specify the information regarding the postoperative follow-up. How were the intervals for follow-up and what kind of examinations or work-ups were carried out.

I am sorry not clearly indicated this issue and, however this was described in Method section. Please confirm it. Each institute may do the similar follow-up interval or means in Japan or Kyushu district (line 181-183)

6) Regarding the patients with HCC, underlying liver diseases, including cirrhosis, hepatitis, and etc., are also important factors, which have impact on perioperative outcomes of the patients. More detailed presentation of these data and subsequent consideration in analyses is needed.

I am sorry for not indicating well in the initial manuscript. However, this was already indicated in Table 1 and 2. In the part of histological findings, fibrotic and necroinflammatory (detail information of background liver injury) degree was indicated. To understand easily, this was classified as normal/ chronic hepatitis/ cirrhosis by the final histological diagnosis, which were indicated adjacent to F degree both in Table 1 and 2. Moreover, in the patient demographic part of Table1, existence of preoperative mortal hypertensive findings or ascites, and degree of splenomegaly relating to background liver injury were added. However, prevalence of these existences and splenomegaly were quite less and these could not be analyzed with posthepatectomy complications. This point was slightly described in Discussion (line 372-378)

7) Please provide the tumor location as well as type of mesohepatectomy based on the classification provided by Qiu et al. (Surg Oncol. Qui et al., 2016 Dec;25(4):441-447. doi: 10.1016/j.suronc.2016.03.001.)

We appreciate the reviewer’s important suggestion. As the reviewer’s suggestion, we referred this useful classification and add information with analysis as new reference (13). The classification was explained in Method section (151-157). This results were indicated in Table 1 and 2. Type IV was significantly associated with PHLF and type I was associated with BL, and usefulness of this simple classification was clarified in CB. This classification and other parameters were statistically confounding and this classification included various influences of HCC to vessels as abutment, compression and invasion. Therefore, this classification was only applied for the univariate analysis but not multivariable analysis in this study. This results and description were described in Results (line 253-264 and 284-285) and Discussion (line 444-452).

8) Many variables and their categorization are not mentioned in methods and materials. For instance, the references for categorization of tumor distance to vascular structures and their involvement, should be provided.

In the Method section, references were applied as well as possible. With respect to other parameters as distance between HCC and vessels were our original idea to evaluate the influences for operative risks. This was described in Discussion (line 452-461).

9) It should also be reported, how many vascular resections were carried out, if any. Besides, tumor compression and tumor infiltration to major vessels are two completely different entities, which cannot taken into account similarly, since intraoperative approaches to these conditions are totally different.

In the present series, eight patients underwent combined vascular resections, 1 for partial resection of IVC, 2 for right hepatic veins (1 partial resection and 1 graft reconstruction) and 5 for hepatic duct (hepaticojejunostomy). Basically, compression of HCC was supposed to be dissected because of characteristics of morphology and then, surgeons in the present series have attempted to challenge CB in case of compressive HCC. As the reviewer, however, patterns of vascular involvements were various as the Qiu’s classification in centrally located HCC. This problematic point was described in the Results (line 284-285) and Discussion (line 447-456), respectively.

10) Single patient had an intraoperative bloods loss of 23L and duration of OP was 30 hours. Please provide further details regarding this case. 

We did not described operation time in a mortal patient as Fig 3 and Table 5. This patient died within 30 days after surgery (actually day 26 due to progressive PHLF). Intraoperative blood loss was indeed 23L which was due to transient coagulopathy with bleeding tendency for a hour. Eventually, we could stop bleeding during operation. Operation time was approximately 14 hours. After submission, information of this case was required under acceptance of patient’s family from the journal’s policy. As we cannot receive in these all four cases at this stage, we are sorry but the figure 3 and table 5 must be deleted in resubmission.

11) Please use the standard and well-known abbreviation and terminology for description of the outcomes including PHBL and PHLF. As shown in the figures, the outcomes of Kaplan-Meier analysis should be reported in the Results section or a table in detail.

We are sorry for using unusual terminology. The term of PHBL and PHLF were used in the entire text and Tables. Actual number of patients per year was added as a table in the figure 4.

Reviewer 2 Report

We recommend some changes:
- We believe this article is suitable for publication in the journal although some revisions are needed. The main strengths of this paper are that it addresses an interesting and very timely question and provides a clear answer, with some limitations. Certainly, the study is limited to an Asian population with a relatively small sample size, and authors should further express this point. Thus, the authors should better highlight the limitations of the current paper.

- A linguistic revision is needed.
- The background of the changing scenario of medical treatment in HCC should be better discussed, and some recent papers regarding this topic should be included ( 
PMID: 34976841PMID: 36368251).

Author Response

February 28, 2023

 Dear Editor-in-chief, and editorial office staff

Cancers,

cancers-2229392R1; Manuscript Title: Risk Factors of Complications in Central Bisectionectomy (H458) for Hepatocellular Carcinoma: A Multi-institutional Single-arm Analysis (the title was revised according to reviewer’s comment)

Principle author: Atsushi Nanashima

Dear editor-in-chief,

Please find enclosed the above resubmitted manuscript to Cancers as a full article.  

We are so sorry to know that the present paper has been rejected under your web site comments. According to reviewer’s comments, however, possibility of resubmission may remained and two of three reviewers might potentially accepted our manuscript. One reviewer who denied reviewing also showed the forward opinions in the comment. Thus we have attempted resubmission according to these comments which was revised by point-by-point response by yellow highlighted as the editor mentioned.

After the initial submission, the journal’s office pointed out that the personal information as case series report was shown in Table 5 and Figure 3. These were indicated as representative cases of mortality at first. The office required the acceptance with assignment of family’s name and, however, we cannot contact all four cases actually. Therefore, we would like to request deletion of these data in re-submission manuscript. The authors also understand that they will automatically transfer the copyright to the publisher. We look forward to publication of our revised manuscript. The authors also understand that should be submitted material been accepted for publication in the journal, they will automatically transfer the copyright to the publisher.

I hope that the reviewing process finds the resubmitted manuscript acceptable for publication in the journal.

 Sincerely,

Professor Atsushi Nanashima, M.D., PhD., and FACS.

Division of Hepato-biliary-pancreas surgery, Department of Surgery, University of Miyazaki Faculty of Medicine, Kiyotake 5200, Miyazaki, 889-1692, Japan.

E-mail; a_nanashima@med.miyazaki-u.ac.jpTelephone: +81-985-85-2905;Fax: +81-985-85-3780

 cancers-2229392R1; Manuscript Title: Risk Factors of Complications in Central Bisectionectomy (H458) for Hepatocellular Carcinoma: A Multi-institutional Single-arm Analysis (the title was revised according to reviewer’s comment)

 Response to reviewer 2

We believe this article is suitable for publication in the journal although some revisions are needed. The main strengths of this paper are that it addresses an interesting and very timely question and provides a clear answer, with some limitations.

We are so pleased to know the reviewer’s sincere comments. By the reviewer’s comments, we still believe that our manuscript was potentially accepted. We also appreciate for very careful check and the reviewer gave us useful comments or suggestions. This can be revised. We suppose that resubmission would be reconsidered by the same reviewer’s sincere comment. Revised part was indicated by yellow highlighted accompanied with point-by-point responses as below.

(x) I would like to sign my review report

English language and style

 (x) Moderate English changes required.  English language in the entire text was again edited by the Elsevier English editing service as indicated in line 566-568.

Yes        Can be improved Must be improved             Not applicable

Does the introduction provide sufficient background and include all relevant references?

( )          (x)         ( )          ( )

Are all the cited references relevant to the research?

( )          ( )          (x)         ( )     This was revised according to the reviewer’s comments

Is the research design appropriate?

( )          (x)         ( )          ( )

Are the methods adequately described?

( )          ( )          (x)         ( )    This was revised according to the reviewer’s comments

Are the results clearly presented?

( )          ( )          (x)         ( )    This was revised according to the reviewer’s comments

Are the conclusions supported by the results?

( )          ( )          (x)         ( )    This was revised according to the reviewer’s comments

1) Certainly, the study is limited to an Asian population with a relatively small sample size, and authors should further express this point. Thus, the authors should better highlight the limitations of the current paper.

We appreciate your suggestions because clinical situation or status were influenced by the districts or countries. This comment was added in the Discussion (line 496-499).

2) A linguistic revision is needed.

As described in acknowledgement, the entire text was carefully edited by the world-class editing company, Elsevier and described this issue in the initial manuscript (line 563-564). If the reviewer felt the problem of English language and requires re-editing again, we reluctantly order editing in the revised version (Acknowledgements; line 555-568).

3) The background of the changing scenario of medical treatment in HCC should be better discussed, and some recent papers regarding this topic should be included (PMID: 34976841; PMID: 36368251).

Although the present study is to clarify the operative risk of CB but not examine the oncological point of view, recommended two Italian review references regarding the novel chemotherapy and ICI (new ref 34 and 35) were added in the Discussion part regarding limitation issue in revision (line 517-521).

Reviewer 3 Report

The article is discussing an interesting topic regarding the surgical treatment by central bisectionectomy in HCC through a retrospective multicenter study conducted on 142 patients. Some major comments are to be addressed.

1. The title of the article includes un-necessary words that is better to be deleted such as Specific and Severe, because there is no grades exist of mild or severe complications.

2. Do the authors have explanation why the HCC lesion in figure 2 D is not enhancing although it represents the arterial phase?

3. In results section, line 200: how the authors considered patients to have non-curative HCC? Similarly, in line 203 why the authors considered 2 and 3 lesions of HCC as multiple HCC? although Barcelona Clinic HCC staging consider lesions >3 in number as multiple or multinodular.

4. Add to the heading of Table 1 (laboratory, radiological and pathological characteristics), because it does not illustrate clinical characteristics only as mentioned.

5. In table 1, what does the grading of macrovascular involvement (0 1/2,3) represent? there is also a missing third value (two values only are present in the table).

6. In table 1, what does the abbreviation PVE in preoperative cancer treatment mean?

7. How is the data in Table 2 are represented ie. mean or median or frequency? where is the p-value? the presented categorical data are not-understood at all. Please, re-write this table in a comprehensible form.

8. In results section, sub-title Postoperative patients outcome for three years, line 303; you described the mean duration of hospital stay as 34 +/ 28 days while in table 1 you described it using median and IQR (26[8-176]). Please, correct it in the text as it is a non-parametric data.

9. In results section, in sub-title Postoperative patients outcomes for three years, line 305; the number of patients with postoperative tumor relapse is 71 (51%), while in table 1 it is 71(50%) and the sum of this relapse in the table (liver 59, extra-liver 10, lung 1) equals 70 not 71. Can you explain these differences? 

10. In results section, sub-title Postoperative patients outcome, line 306; you referred to table 1 as 66 patients received postoperative anti-tumor relapse treatment, while in table 1 it is described under the heading "pre-operative cancer treatment". Please, explain.

Similarly, in table 1 the number of cancer-free survival is 109 while in line 307, you mentioned the number to be 67. Again, explain this difference.

11. Did any patient undergo liver transplantation after CS bisectionectomy to treat HCC relapse?

12. In the simple summary, the abbreviation iBL is mentioned without the original word. In the heading of table 5 and figure 3, please correct the word mortal into mortality. 

13. The IQR of prothrombin activity in table 1 is [44-139] and the upper level of normal is 100%, so how did the patient had level above the normal?

Author Response

February 28, 2023

 Dear Editor-in-chief, and editorial office staff

Cancers,

cancers-2229392R1; Manuscript Title: Risk Factors of Complications in Central Bisectionectomy (H458) for Hepatocellular Carcinoma: A Multi-institutional Single-arm Analysis (the title was revised according to reviewer’s comment)

Principle author: Atsushi Nanashima

 Dear editor-in-chief,

Please find enclosed the above resubmitted manuscript to Cancers as a full article.  

We are so sorry to know that the present paper has been rejected under your web site comments. According to reviewer’s comments, however, possibility of resubmission may remained and two of three reviewers might potentially accepted our manuscript. One reviewer who denied reviewing also showed the forward opinions in the comment. Thus we have attempted resubmission according to these comments which was revised by point-by-point response by yellow highlighted as the editor mentioned.After the initial submission, the journal’s office pointed out that the personal information as case series report was shown in Table 5 and Figure 3. These were indicated as representative cases of mortality at first. The office required the acceptance with assignment of family’s name and, however, we cannot contact all four cases actually. Therefore, we would like to request deletion of these data in re-submission manuscript. The authors also understand that they will automatically transfer the copyright to the publisher. We look forward to publication of our revised manuscript. The authors also understand that should be submitted material been accepted for publication in the journal, they will automatically transfer the copyright to the publisher.

I hope that the reviewing process finds the resubmitted manuscript acceptable for publication in the journal.

 Sincerely,

Professor Atsushi Nanashima, M.D., PhD., and FACS.

Division of Hepato-biliary-pancreas surgery, Department of Surgery, University of Miyazaki Faculty of Medicine, Kiyotake 5200, Miyazaki, 889-1692, Japan.

E-mail; a_nanashima@med.miyazaki-u.ac.jpTelephone: +81-985-85-2905;Fax: +81-985-85-3780

 cancers-2229392R1; Manuscript Title: Risk Factors of Complications in Central Bisectionectomy (H458) for Hepatocellular Carcinoma: A Multi-institutional Single-arm Analysis (the title was revised according to reviewer’s comment)

 Response to reviewer 3

 The article is discussing an interesting topic regarding the surgical treatment by central bisectionectomy in HCC through a retrospective multicenter study conducted on 142 patients. Some major comments are to be addressed.

(x) I would like to sign my review report

English language and style

(x) I don't feel qualified to judge about the English language and style

Yes         Can be improved Must be improved             Not applicable

Does the introduction provide sufficient background and include all relevant references?

(x)          ( )          ( )          ( )

Are all the cited references relevant to the research?

(x)          ( )          ( )          ( )

Is the research design appropriate?

(x)          ( )          ( )          ( )

Are the methods adequately described?

(x)          ( )          ( )          ( )

Are the results clearly presented?

(x)          ( )          ( )          ( )

Are the conclusions supported by the results?

(x)          ( )          ( )          ( )

We are pleased to know that the reviewer’s useful comment and the best evaluation in the check points, which may allow our manuscript potentially accepted. We also appreciate the additional comments to improve the quality of manuscript.

  1. The title of the article includes un-necessary words that is better to be deleted such as Specific and Severe, because there is no grades exist of mild or severe complications.

We really appreciate the reviewer’s instruction, which was previously recommended. However, as the reviewer suggested, this unnecessary words were deleted in the title and abstract (line 2 and 21).

  1. Do the authors have explanation why the HCC lesion in figure 2 D is not enhancing although it represents the arterial phase?

 This figure 2D demonstrates the distances between HCC and vessels, but not intra-tumorous characteristics. This CT figure was image at the arterio-portal phase (not early arterial phase) and, therefore, tumor parenchyma showed a characteristics of wash-out appearance in compared to the normal liver parenchyma, which is also a specific characteristics of classical HCC. By noting carefully, some nodules in HCC still showed remnant enhancement. The representative adequate figures could not be obtained actually. Please realize it.

 3. In results section, line 200: how the authors considered patients to have non-curative HCC? Similarly, in line 203 why the authors considered 2 and 3 lesions of HCC as multiple HCC? Although Barcelona Clinic HCC staging consider lesions >3 in number as multiple or multinodular.

We are sorry for applying the wrong expression as “non-curable HCC treatments”, which was revised as “non-surgical HCC treatments” (line 219). We mean TAE or other prior treatments. In addition, we showed the wrong number of this in Table 1. The number of TAE was 15 (11%) in the revised Table 1. In the present study, as indicated as Japan clinico-pathological classification (reference 12) (shown in the line 139-163). In the study of advanced stage HCC (Milan criteria non-met) or unresectable HCC, we recently use BCLC criteria and, however in this study, we define the number of tumor as solitary versus multiple (2 or more). Please realize our manner.

  1. Add to the heading of Table 1 (laboratory, radiological and pathological characteristics), because it does not illustrate clinical characteristics only as mentioned.

Headline title of table 1 was revised as reviewer’s instructions (line 234).

  1. In table 1, what does the grading of macrovascular involvement (0 1/2,3) represent? there is also a missing third value (two values only are present in the table).

We applied the Japan classification as reference 12 as described above, and we combined number of grade 0 and 1 vs. 2 and 3. However, it seemed to be wrong and lead misunderstanding meaning of this parameter. Thus, this was revised as no / yes. The explanation indicating § was annotate in the footnote of table 1 (line 244-245). Value was tidy two in each.

  1. In table 1, what does the abbreviation PVE in preoperative cancer treatment mean?

We are sorry the unnecessary procedure of portal vein embolization in Table 1, which was deleted in revision. This was a mistake.

  1. How is the data in Table 2 are represented ie. mean or median or frequency? where is the p-value? the presented categorical data are not-understood at all. Please, re-write this table in a comprehensible form.

As indicated in the Methods and footnote, continuous data were unified as mean +/- SD. We applied the Scheff’s Multiple Comparison Procedure for statistical analysis. Significant different data were indicated as *p<0.05 and * p<0.01 as footnotes. As the reviewer’s useful comments, the revised combined table II with continuous and categorical data showed so much better. We appreciate the reviewer so much!

  1. In results section, sub-title Postoperative patients outcome for three years, line 303; you described the mean duration of hospital stay as 34 +/ 28 days while in table 1 you described it using median and IQR (26[8-176]). Please, correct it in the text as it is a non-parametric data.

The median value was added in the revised manuscript (line 322-323) although non-parametric.

  1. In results section, in sub-title Postoperative patients outcomes for three years, line 305; the number of patients with postoperative tumor relapse is 71 (51%), while in table 1 it is 71(50%) and the sum of this relapse in the table (liver 59, extra-liver 10, lung 1) equals 70 not 71. Can you explain these differences?

We are so sorry for discordance of tumor relapse number and its rate. These were checked again and revised as Table 1 and text (line 326). This error might happened by the calculation with or without four patients of in-hospital death.

  1. In results section, sub-title Postoperative patients outcome, line 306; you referred to table 1 as 66 patients received postoperative anti-tumor relapse treatment, while in table 1 it is described under the heading "pre-operative cancer treatment". Please, explain.

We apologize that our data might confuse the reviewer. In table 1, non-surgical treatments before CB was only indicated and we missed to indicate in the text. Position of the word (Table 1) was moved. Please realize it.

  1. Did any patient undergo liver transplantation after CS bisectionectomy to treat HCC relapse?

No salvage liver transplantation was undergone in this series. Not describe this issue in the revision.

  1. In the simple summary, the abbreviation iBL is mentioned without the original word. In the heading of table 5 and figure 3, please correct the word mortal into mortality.

In the conclusion, iBL was changed as line 553. After the first submission, the journal’s office pointed out that the table 5 and figure 3 were case series information. Although the patient family’s assignment for these information was required, it could not be obtained because we could not contact all. Therefore, it is unfortunate but these results must be deleted according to the journal’s policy.

  1. The IQR of prothrombin activity in table 1 is [44-139] and the upper level of normal is 100%, so how did the patient had level above the normal?

The normal standard level is theoretically 100% and, in fact, increased level of PT activity was sometimes observed in patients with chronic liver diseases. In this series, 30 patients (21%) showed over 100% including 130% level in two patients and 120% level in two patients. I confirmed the laboratory expert and hematological physician that the normal upper limit of prothrombin activity was actually ranged between 130 and 139% in institutional upper range in Japan. While, the lower normal limit was mostly set at around 80%. The PT less than 80% may indicate clinically abnormal coagulant state, particularly in chronic liver injury diseases, which is so useful parameter reflecting liver functions. We are sorry but the clinical significance of increased PR activity over 100% may not been clarified by consulting any expert physicians at this stage.
